# A photoheterotrophic bacterium from Iceland has adapted its photosynthetic machinery to the long days of polar summer

Jürgen Tomasch,[1] Karel Kopejtka,[1] Tomáš Bílý,[2] Alastair T. Gardiner,[1] Zdenko Gardian,[2] Sahana Shivaramu,[1] Michal Koblížek,[1] David Kaftan[1,3]

**ABSTRACT** During their long evolution, anoxygenic phototrophic bacteria have inhabited a wide variety of natural habitats and developed specific strategies to cope with the challenges of any particular environment. Expression, assembly, and safe operation of the photosynthetic apparatus must be regulated to prevent reactive oxygen species generation under illumination in the presence of oxygen. Here, we report on the photoheterotrophic *Sediminicoccus* sp. strain KRV36, which was isolated from a cold stream in north-western Iceland, 30 km south of the Arctic Circle. In contrast to most aerobic anoxygenic phototrophs, which stop pigment synthesis when illuminated, strain KRV36 maintained its bacteriochlorophyll synthesis even under continuous light. Its cells also contained between 100 and 180 chromatophores, each accommodating photosynthetic complexes that exhibit an unusually large carotenoid absorption spectrum. The expression of photosynthesis genes in dark-adapted cells was transiently downregulated in the first 2 hours exposed to light but recovered to the initial level within 24 hours. An excess of membrane-bound carotenoids as well as high, constitutive expression of oxidative stress response genes provided the required potential for scavenging reactive oxygen species, safeguarding bacteriochlorophyll synthesis and photosystem assembly. The unique cellular architecture and an unusual gene expression pattern represent a specific adaptation that allows the maintenance of anoxygenic phototrophy under arctic conditions characterized by long summer days with relatively low irradiance.

**IMPORTANCE** The photoheterotrophic bacterium *Sediminicoccus* sp. KRV36 was isolated from a cold stream in Iceland. It expresses its photosynthesis genes, synthesizes bacteriochlorophyll, and assembles functional photosynthetic complexes under continuous light in the presence of oxygen. Unraveling the molecular basis of this ability, which is exceptional among aerobic anoxygenic phototrophic species, will help to understand the evolution of bacterial photosynthesis in response to changing environmental conditions. It might also open new possibilities for genetic engineering of biotechnologically relevant phototrophs, with the aim of increasing photosynthetic activity and their tolerance to reactive oxygen species.

**KEYWORDS** AAP, gene expression, light adaptation, photosynthesis, Proteobacteria, *Sediminicoccus*

Almost all the metabolic energy that powers life on this planet originates directly or indirectly from solar radiation. One of the first groups to evolve the ability to harvest light energy was anoxygenic phototrophic Proteobacteria (1). These organisms harvest light using bacteriochlorophyll (BChl) and carotenoid molecules bound to photosynthetic (PS) complexes (2). Phototrophic Proteobacteria evolved under anaerobic conditions during the Archean eon (3). With the emergence of Cyanobacteria 2.4 billion years ago (4), the concentration of atmospheric oxygen gradually rose during the entire

Address correspondence to David Kaftan, kaftan@alga.cz.

The authors declare no conflict of interest.

See the funding table on p. 14.

Proterozoic Eon until it reached the current levels approximately 0.54 billion years ago (5). BChl synthesis represents a major challenge for anoxygenic phototrophs in the modern oxic world. The membrane-bound intermediates of BChl synthesis, when illuminated in the presence of oxygen, can generate dangerous reactive oxygen species (ROS) (6, 7). The long evolutionary transition led to a wide differentiation of phototrophic proteobacterial lineages that developed different mechanisms for safely expressing, assembling, and operating their PS apparatus. Some lineages (e.g., purple sulfur bacteria) remained anaerobic and retreated to remaining anoxic environments (8). Other groups, such as purple non-sulfur bacteria, developed an intricate redox regulatory system repressing their pigment synthesis in the presence of oxygen (9, 10), making it possible to conduct photosynthesis in micro- or semiaerobic habitats.

The last group of phototrophic species fully adapted to the oxic atmosphere and became obligately aerobic. These so-called aerobic anoxygenic phototrophic (AAP) bacteria are a common part of the aquatic bacterioplankton inhabiting euphotic zones of rivers, lakes, and oceans (11, 12). They are typically photoheterotrophs with a significantly reduced light-harvesting apparatus compared to their anaerobic relatives (13). To avoid the risk of ROS generation under aerobic conditions (14), AAP bacteria studied so far rapidly repress photosynthesis gene expression when illuminated, restricting BChl synthesis to the night (15–20). This kind of regulation has also been observed in natural environments with an alternating light/dark regime throughout the year (21–23). The main components of the photosynthesis regulatory system are conserved among all three types of anoxygenic phototrophs. The common aerobic repressor-antirepressor system relies on the redox-responsive transcription factor PpsR, which is central to the light intensity and oxygen concentration-dependent repression of the photopigment biosynthesis genes (24–26).

As phototrophic Proteobacteria have expanded to almost all sun-lit environments on Earth, including the vast areas of polar regions (27), they have had to adapt to a variety of light regimes. Here, we cultured and characterized a BChl-producing bacterium KRV36 isolated from an oligotrophic stream near Raufarhöfn in north-western Iceland, 30 km south of the Arctic Circle. This area experiences almost no dark periods during the summer. In continuous light, the strain remained pigmented with the continuous presence of chromatophores and high PS activity. To understand the strain's adaptation to the polar light conditions, we analyzed the photosystem structure and function, sequenced the genome, and recorded the transcriptional response of this arctic AAP bacterium to changes in illumination.

## RESULTS

### Isolation, identification and genome sequencing of KRV36

Half-strength Reasoner's 2A (R2A) agar plates were inoculated with serially diluted samples collected in northern Iceland near Raufarhöfn in July 2020 (Fig. 1A). Samples contained homogenized cyanobacterial mats (Fig. 1B). BChl-containing colonies were identified using IR fluorescence screening and further purified by passaging to receive single cell colonies (Fig. 1C). The isolated strain KRV36 grown aerobically in liquid 1/2 R2A medium formed small coral-red aggregates with a radius of 1 mm that adhered into larger mesh-like, spherical clusters with diameter up to 4 cm (Fig. S2). The cells grew optimally at 23°C. No growth was observed above 30°C. Interestingly, the KRV36 cultures produced BChl under both light and dark conditions (Table 1). The identity of KRV36 was inferred using 16S rRNA phylogeny, documenting a close relationship to *Sediminicoccus rosea* R-30$^T$ (99.2% pairwise 16S rRNA sequence similarity, Fig. 1D). The genome of KRV36 (NCBI GenBank accession number: GCA_023243115.1) consists of a single 4.91 Mb chromosome with 2 rRNA gene clusters, 49 tRNAs, and 4,576 protein-coding genes. Genome characteristics are summarized in Table S2. The chromosome contains two copies of the rRNA operon and also two remnant and two almost complete phage genomes (Fig. S1). Genes for sugar utilization using glycolysis/gluconeogenesis and the pentose phosphate pathway, TCA cycle, and biosynthetic pathways for all proteinogenic

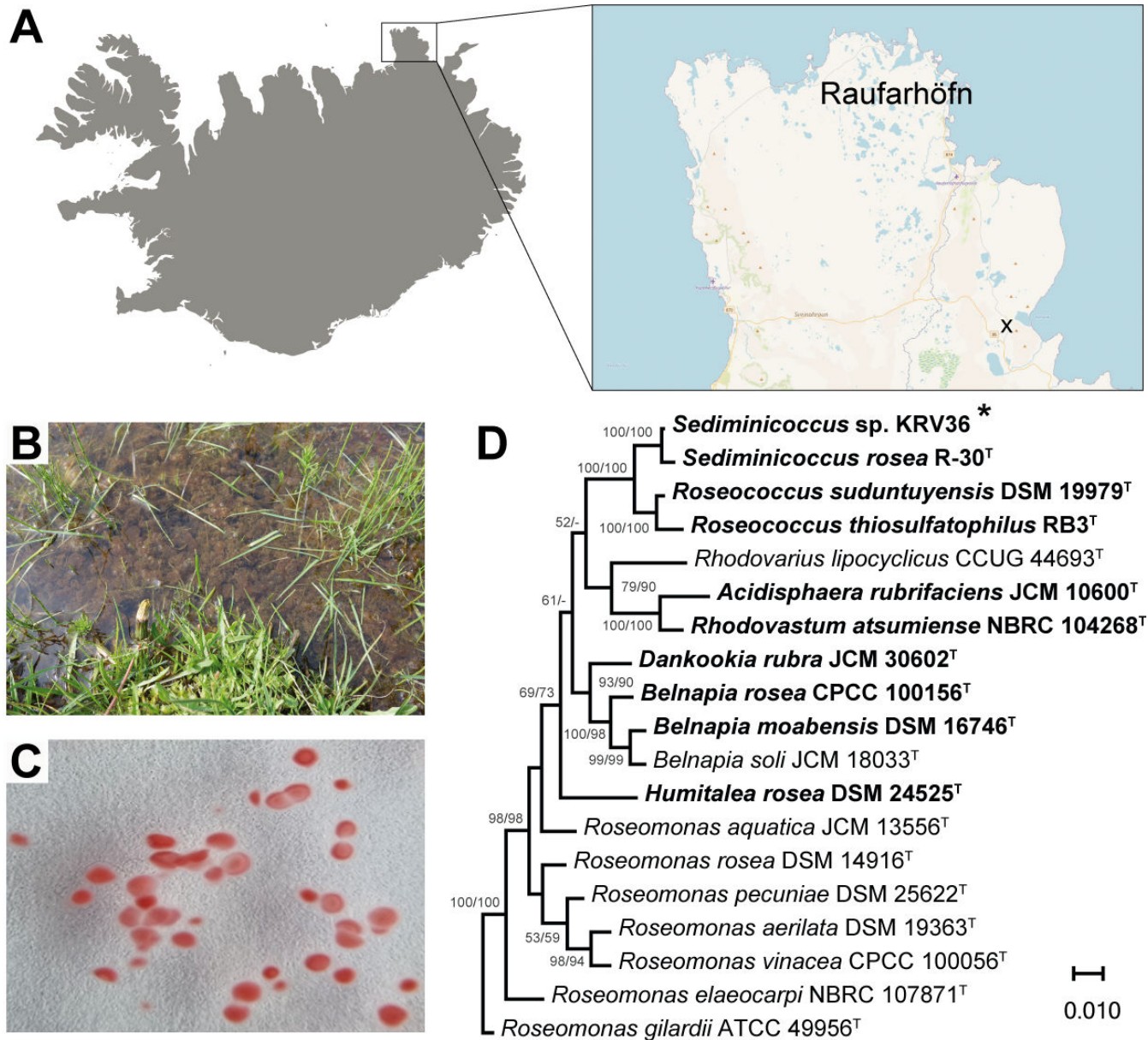

**FIG 1** Isolation and identification of *Sediminicoccus* sp. KRV36. (A) Position of sample collection in northern Iceland in the littoral of the right tributary of the Kollavíkurá stream above the valley of Lake Kollavíkurvatn is marked by a cross at 66°16′22.4″N 15°48′24.5″W. (B) Cells were isolated from the mats of nostocacean cyanobacteria. (C) Cells grown on half-strength R₂A medium agar formed small colonies with a radius of 1 mm. (D) 16S rRNA phylogenetic tree showing the position of the KRV36 strain (marked with an asterisk) within the family *Acetobacteraceae*. *Roseomonas gilardii* was used as an outgroup organism. Scale bar represents amino acid changes per position. Neighbor-joining/maximum likelihood bootstrap values > 50% are shown. Species with PGC are in bold. Maps were obtained from Free Vector Maps (https://freevectormaps.com/) and Mapy.cz (Seznam.cz, a.s., https://www.seznam.cz/nastaveni-souhlasu/).

amino acids are present. The strain possesses metabolic potential for gaining nitrogen through assimilatory nitrate reduction or urease. Genes coding for enzymes capable of sulfite reduction and thiosulfate oxidation are also present in the chromosome. Phosphate and phosphonate are imported by ABC transporters. The genome contains both low- and high-$O_2$ affinity cytochrome c oxidases for the final step of oxidative phosphorylation. The genes for BChl and carotenoid biosynthesis and photosystem subunits are present in one photosynthesis gene cluster (PGC) and an additional *puc* operon for the peripheral light-harvesting proteins. The general structure of the PGC resembles structures common in all aerobic anoxygenic photosynthetic bacteria (28),

**TABLE 1** Photosynthetic activity of the RC-LH1 in *Sediminicoccus* sp. KRV36 cells[a]

|  | Dark | Light |
| --- | --- | --- |
| $\Phi_{\text{RC-LH1}}$ | $0.63 \pm 0.01$ | $0.62 \pm 0.01$ |
| $\sigma_{\text{RC-LH1}}$ | $115 \pm 5$ Å | $106 \pm 2$ Å |
| $J_{\text{con}}$ | $1.55 \pm 0.03$ | $1.52 \pm 0.01$ |
| $k_{\text{re-open}}$ | $2{,}175 \pm 252$ s$^{-1}$ | $2{,}160 \pm 706$ s$^{-1}$ |
| BChl $a$/proteins (wt:wt) | $2.61 \pm 1.90 \times 10^{-3}$ | $1.68 \pm 1.03 \times 10^{-3}$ |

[a]Dark, cells grown in dark for 3 days; light, dark-adapted cells grown in light for 1 day; $\Phi_{\text{RC-LH1}} = F_V / F_M$, maximum photochemical yield; $\sigma_{\text{RC-LH1}}$, optical cross-section; $J_{\text{con}}$, Joliot's connectivity; $k_{\text{re-open}}$, RC reopening rate; and BChl a/proteins, bacteriochlorophyll to protein (wt:wt) ratio.

although with several additional unique deviations. Notably, an additional gene for heme synthesis is present. Genes of two regulators, *ppaA* and *ppsR*, coding for an antirepressor-repressor system that utilizes a redox-responding protein PpsR responsible for light intensity and oxygen concentration-dependent repression of the *puc* operon and photopigment biosynthesis genes, are at the end of the operon, preceded by half of the bacteriochlorophyll synthesis genes (*bchFNBHLM*). Genes for RubisCO are absent. Other major metabolic pathways are summarized in Tables S3 and S4.

## Cell morphology

A longitudinal TEM cross-section of the elongated cells with a length of 1.8–2 µm and a width of 0.5–0.6 µm showed the cell pole-proximal space filled with intracytoplasmic membrane (ICM) vesicles with a diameter of $106 \pm 13$ nm (Fig. 2A) resembling the chromatophores of purple non-sulfur phototrophic bacteria, e.g., *Rhodobacter* species. Cell walls were decorated by a structured mucilaginous layer. Analysis of TEM and atomic force microscopy (AFM) imaging (Fig. S3) provided a mean width of $28.4 \pm 6.2$ nm and length of $35.3 \pm 8.6$ nm of the individual spikes ($n = 49$) at the surface of the cells. Fluorescence microscopy imaging of DAPI-stained cells showed a BChl fluorescence signal at both cellular poles (Fig. 2B) that coincided with the localization of the ICM vesicles. The central region of the cell exhibited maximal intensities of the DAPI fluorescence (Fig. 2B) also in line with the EM images. The EM tomography analysis guided by the fluorescence imaging revealed a polar arrangement of the BChl containing ICM vesicles in the non-dividing cell (Fig. 2C) and during the process of cell division (Fig. S4).

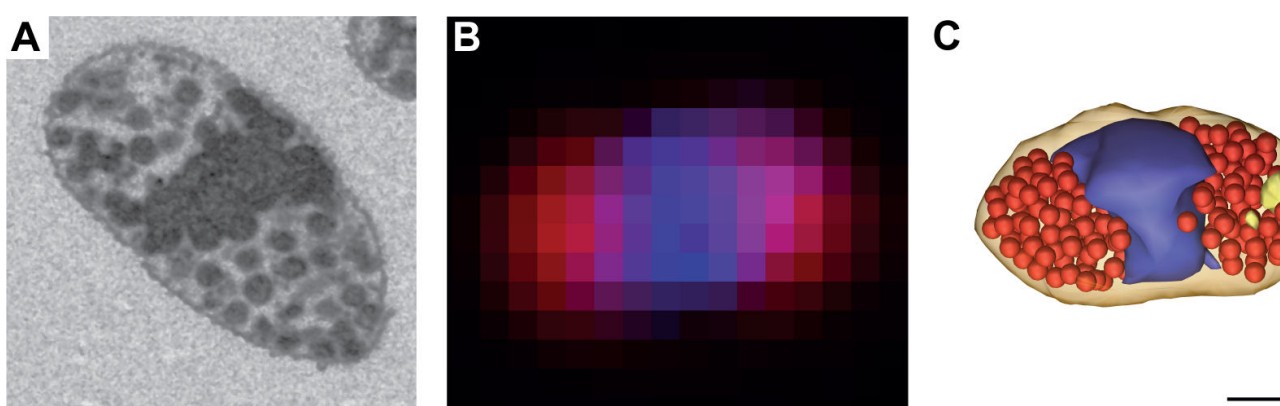

**FIG 2** Imaging of *Sediminicoccus* sp. KRV36 cell ultrastructure. (A) Array tomography (scanning electron microscopy) of one slice of intracytoplasmic vesicles and chromatophores, concentrated at the cell poles. (B) Fluorescence light microscopy image of the overlap between BChl fluorescence and stained DNA. (C) Correlative imaging of the ultrastructure of KRV36 cells by electron microscopy visualizes the arrangement of chromatophores (red) and storage compound granules (beige) and outlines the central vesicle-free space containing DNA (blue). All images are scaled to the same dimensions; scale bar represents 500 nm.

## PS apparatus and activity

The *in vivo* absorption spectrum of KRV36 cells displayed a major infrared BChl *a* band at 871 nm originating from the Qy absorption band of BChl *a* molecules bound to LH1 complexes (Fig. 3A). The second small infra-red absorption peak at 798 nm probably originated from the outer antenna LH2. An intense absorption of carotenoid in the blue part of the spectrum (Fig. 3A), with the main absorption peaks at 483, 508, and 540 nm, was responsible for most of the absorption between 400 and 600 nm. The BChl *a* content per protein (Table 1) corresponds to values reported for related *Erythrobacter* species (29). The corresponding number of ~40,000 RC-LH1 complexes per KRV36 cell is slightly above the range of values of 150–30,000 RC-LH1 per cell in a representative sample of AAP bacteria and purple non-sulfur bacteria, respectively (13).

The fluorescence emission spectrum showed a clear peak at 883 nm. The large overlap between the absorption and emission spectrum peaks, respectively (Fig. 3A), suggested a high efficiency of energy transfer between the LH1 complex and RC. The yield of primary photochemical reactions measured by the kinetics of BChl fluorescence (Fig. 3B) in live cells remained high ($F_V/F_M$ = 0.62–0.63) along with high connectivity of the RC-LH1 complexes $J_{con}$ = 1.52–1.55, irrespective of light or dark growth conditions (Table 1). The presence of carotenoids in the LH1 antennae additionally contributes to the exceptionally large optical cross-section of the RC-LH1 complex at 460 nm, $\sigma_{RC-LH1}$ = 106 Å². Sucrose density gradient centrifugation of the solubilized PS membranes revealed an excess of these carotenoids in ICM vesicles alongside the purified RC-LH1 complexes (Fig. 3C). The structural organization of the purified RC-LH1 complex was revealed by cryo-EM in a low-resolution electron density map (Fig. 3D) and denaturing SDS-PAGE displaying the individual proteins of the RC (H, L, M, and C) and the LH1 (α, β) complexes, respectively (Fig. 3E).

Cells showed respiration rate dependence on incident light intensity (Fig. S5), dropping from 6.82 ± 0.26 mg $O_2$ $L^{-1}$ $hour^{-1}$ in the dark down to 2.89 ± 0.24 mg $O_2$ $L^{-1}$ $hour^{-1}$ at 0.9 mmol photon $m^{-2}$ $s^{-1}$. At the saturating light intensities, energy derived from captured light constituted almost 60% of the energy derived from oxidative phosphorylation in the dark.

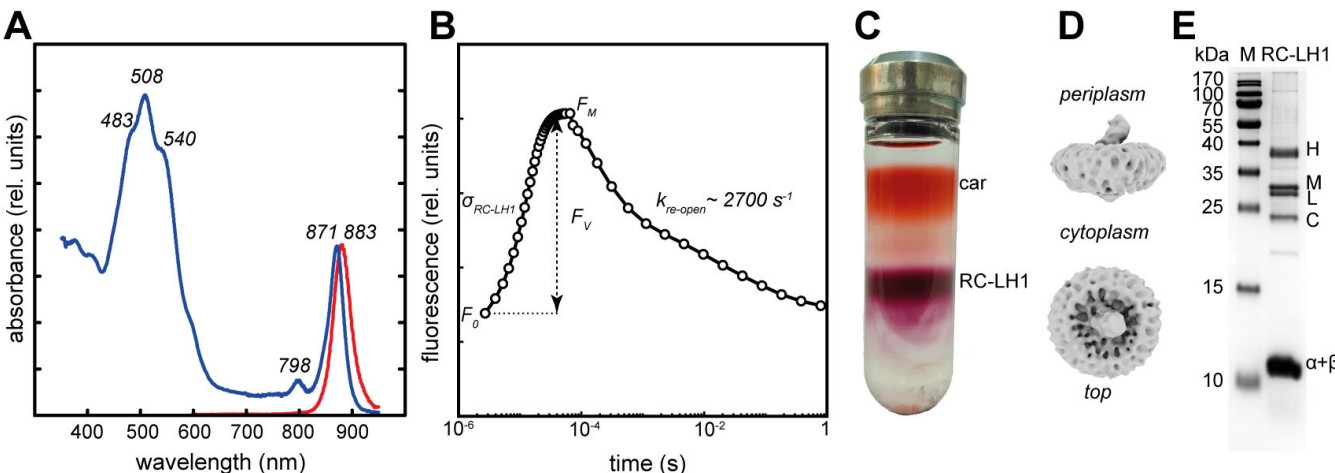

**FIG 3** Physical properties of *Sediminicoccus* sp. KRV36 photosystems. (A) Light absorption (blue) and fluorescence emission (red) spectra of live cells measured at room temperature exhibit signature peaks (absorbance $\lambda_{MAX}$ = 871 nm, $\lambda_2$ = 798 nm, $\lambda_3$ = 540 nm, $\lambda_4$ = 508 nm, $\lambda_5$ = 483 nm, emission $\lambda_{MAX}$ = 883 nm) of photosynthetic complexes. (B) BChl fluorescence induction-decay kinetics of live cells grown at constant light reveals high photochemical yield ($\Phi_{RC-LH1}$ = $F_V/F_M$ = 0.65), large optical cross-section ($\sigma_{RC-LH1}$ = 106 Å), and rapid catalytical turnover (RC reopening rate $k_{re-open}$ = 2,685 $s^{-1}$) of the RC-LH1 complexes, respectively. (C) Sucrose density gradient of the solubilized ICM vesicles showing the upper orange band of membrane carotenoids (car) and the purple band containing RC-LH1 complex. (D) Cryo-EM electron density maps of purified RC-LH1 complexes showing the side view (top) and top view (bottom). (E) Proteins of the RC (H, L, M, and C) and the LH1 antenna (α and β) separated using SDS-PAGE.

## Expression of PS during light regime transition

Cells grown under continuous illumination were still pigmented although with approximately 35% lower Bchl content when compared to the cells grown in the dark (Table 1). To investigate the origin of this difference, we analyzed the transcription of *pufM* gene (encoding the M subunit of the PS reaction center) using reverse transcription quantitative PCR (RT-qPCR). The expression levels of *pufM* (LHU95_13885), a gene coding for the key protein of the RC-LH1 complex, were similarly high and indistinguishable between the light- and dark-grown cells (Fig. 4). Upon the transition from light to dark, the *pufM* gene continued to be constitutively expressed, with expression levels not differing significantly (Tukey's test, *P*-value = 0.75) from the light-adapted levels. A very different pattern was observed in the dark-adapted cells suddenly illuminated by 100 µmol photon $m^{-2}$ $s^{-1}$. Two hours after the dark-light transition, *pufM* expression was lowered significantly (Student's *t*-test, *P*-value = 0.03) to a minimum corresponding to 12% of the initial expression level. Then, the fast repression of the *pufM* expression was alleviated, resulting in a linear increase of *pufM* gene expression ($R^2$ = 0.96, $k$ = 0.39 $\Delta C_t$ $hour^{-1}$) reaching its maximum after 20 hours with 83% of the dark-adapted expression levels.

## Transcriptome dynamics during changing light regimes

To reveal the effects of light on the whole transcriptome, the cultures were first adapted to continuous illumination and darkness for 72 hours. Following the control sampling (time point: 0 hours), the light regime was reversed and cells from both treatments were harvested at 2, 4, 8, and 24 hours after the change. Initially, no significantly differentially expressed genes were obtained for the whole data set (Fig. S6A). Analyzing each sample separately revealed a high variability of expression for a subset of genes, for time points 8 hours after the shift to light and 24 hours after the shift to dark (Fig. S6B). These genes, including the whole PGC, showed transient repression early in the light but different

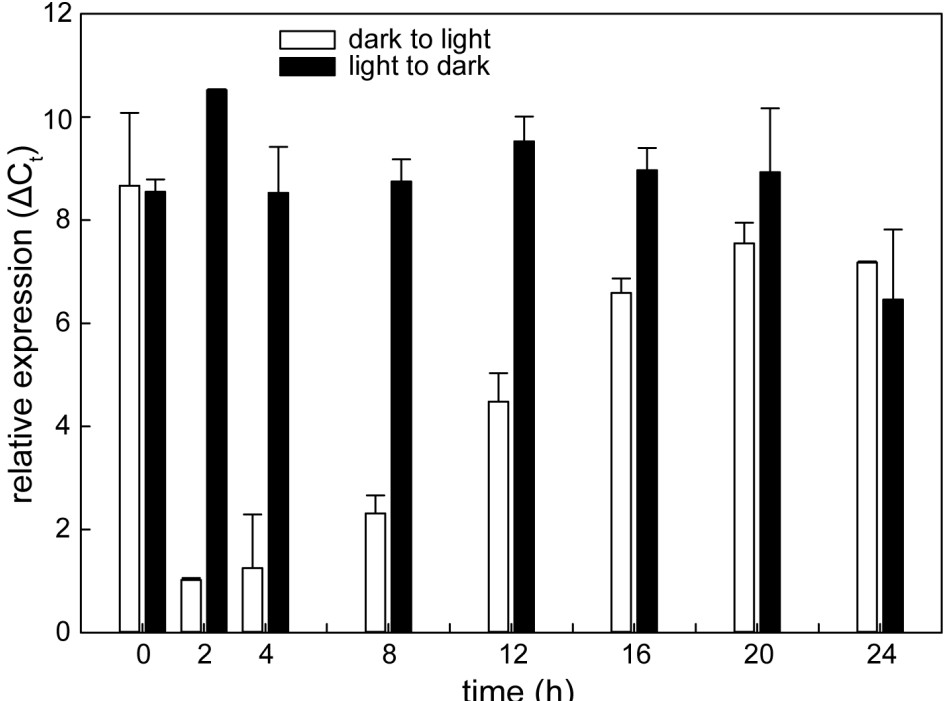

**FIG 4** Transcriptional response of the *pufM* gene to changing light regimes. White bars represent time series for the transition of dark-adapted cells to a continuous-light regime, black bars represent a time series for the transition of light-adapted cells to a continuous-dark regime. The $\Delta C_t$ mean values and standard deviations from two biological replicates are shown. $C_t$, threshold cycle.

recovery times for both samples after 8 and 24 hours in the light. Likewise, only for one sample, a reduction in gene expression was observed after 24 hours in the dark (Fig. S6B). These data are principally in accordance with the qPCR data. In the following, we will consider only genes with a significant change in expression during the first 4 hours after the shift but will show the whole time series keeping variances at later time points in mind. The full data set can be found in Table S5.

In total, 506 genes increased and 499 genes decreased in expression during the first 4 hours of light exposure. The reverse shift from light to dark showed a weaker response with only 54 genes upregulated and 20 genes downregulated (Fig. 5A). Most expression changes for both time series were only transient, and genes reached their base level again at 24 hours (Fig. 5B). Two operons coding for sulfur metabolism genes showed an exceptionally high activity throughout the growth in light with a peak of 12-fold upregulation at 4 hours. Their expression showed a similar but fourfold weaker increase during the shift from dark to light (cluster 1). An operon coding for a phosphate uptake transporter strongly upregulated only 2 hours after the shift to light was part of a small immediate response (cluster 2). Only 14 poorly characterized genes were repressed for the whole 24 hours in light (cluster 3). One cluster of genes, mainly the PGC, showed an immediate 60-fold downregulation up to 4 hours in the light, followed by a full recovery at 24 hours. This cluster also showed a transient fourfold upregulation after the shift to darkness (cluster 4).

Mainly anabolic processes such as glycolysis, amino acid biosynthesis, and the ribosomal proteins were transcriptionally upregulated in the light (Fig. 5C). Genes coding

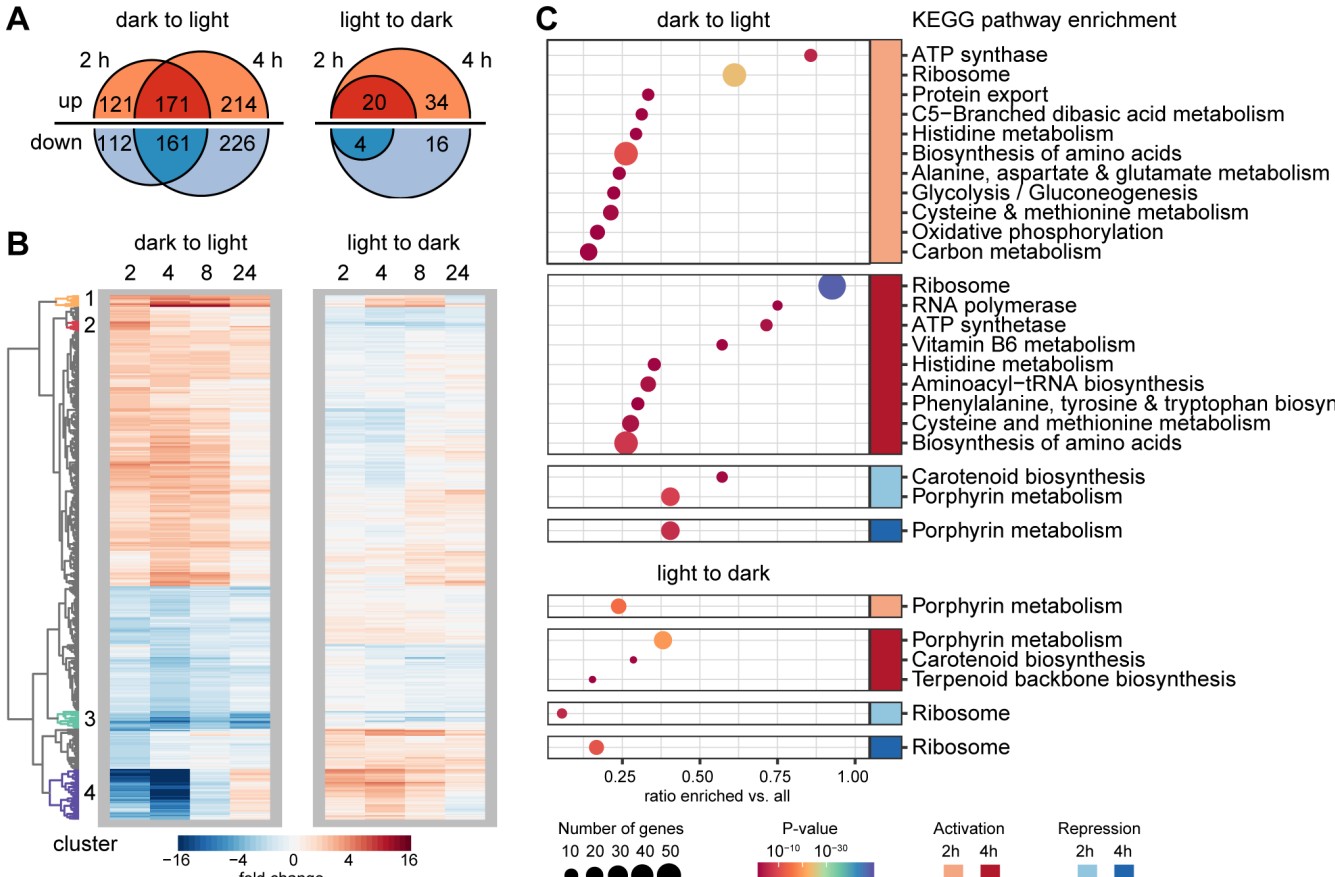

**FIG 5** Response of the transcriptome to changing light regimes. (A) Number of up- and downregulated genes for the first two time points after the transition from dark to light and *vice versa*. No significant changes in expression were detected at later time points. (B) Expression dynamics in both time series for all genes significantly regulated at least at one time point. Four clusters of genes discussed in the text are highlighted. Note that an additional fold change cutoff (±2) was employed for better visualization. (C) KEGG categories significantly enriched during the transition from dark to light and *vice versa*.

for components of the oxidative phosphorylation chain and the ATP synthase were also activated in the light. Of note is also the higher expression of the vitamin B6 biosynthesis pathway. This vitamin is a cofactor of several enzymes involved in redox reactions, in particular, in amino acid biosynthesis (30). Among the downregulated genes, only the BChl and carotenoid pathways were found to be considerably downregulated. The shift from light to dark displayed, to a certain extent, a reversal in the biosynthesis of the pigments and ribosomes enriched among the up- and downregulated genes, respectively. In summary, these data suggest that the utilization of light energy by photosynthesis frees resources for cell growth.

### Expression of photosynthesis-related genes

### Oxidative stress response

Despite the expected higher risk of oxidative stress upon light exposure, we saw no transcriptional changes in the oxidative stress response genes under changing illumination. However, they were among the most strongly expressed genes in all samples from both time series (Fig. 6A). Stress response-specific *rpoH* was the most abundant sigma factor in the transcriptome. It was followed by the catalase and four chaperone genes. Additionally, 6 thioredoxin and 16 glutathione turnover genes showed expression above the median of the transcriptome. The stress-related sigma factor *ecfG* and the fasciclin gene, coding a cell aggregation protein, were not expressed.

### Porphyrin metabolism

BChl is synthesized via a common porphyrin pathway, which is also shared with heme and cobalamin biosynthesis. KRV36 pathway for cobalamin biosynthesis is not complete, but it produces siroheme from uroporphyrinogen III. Siroheme is a cofactor of the sulfite and nitrite reductase genes (Fig. 6B). The *hemA* gene at the entry point to the pathway and one of two *hemC* copies involved in the synthesis of uroporphyrinogen are part of the PGC and were temporarily downregulated in light. While the pathway to heme was not differentially regulated, the genes for siroheme biosynthesis, *cobA* and *cysG1,* were strongly upregulated in light. Both are present in an operon, together with one copy of the sulfite reductase and a peroxiredoxin. The second copy of *cysG*, located in an operon with nitrite and additional sulfite reductase gene, was not differentially regulated. This indicates that upon illumination, the porphyrin pathway may be redirected more toward the biosynthesis of siroheme at the expense of BChl and heme biosynthesis (Fig. 6B).

### Photosynthesis

The genes in the PGC and the distantly located LH2-coding *puc* operon were among the most strongly expressed genes in the dark-adapted cultures (Table S4). After the light exposure, they all showed a similar transient downregulation (Fig. 6C), as was shown before for *pufM* using RTqPCR (Fig. 4). Notably, the regulatory genes *ppaA* and *ppsR* were repressed or activated with the same dynamics and to the same extent as their supposed target genes. The expression pattern of *ppsR* was confirmed in an independent RT-qPCR experiment with higher resolution for the later time points (Fig. S7). We could only detect two binding sites for PpsR in the promoter of the operon starting at *bchC* and one in the promoter of the *puc* operon (Fig. 6C).

## DISCUSSION

*Sediminicoccus* sp. KRV36 was isolated from a cold stream in northern Iceland. The subarctic environment is characterized by a relatively short but intense summer periods with cool temperatures, long days, and only short nights. KRV36 employed several adaptations of its PS machinery to face the challenges of growing in this habitat. Namely, KRV36 cells have BChl content at the higher end of other AAP species (13). Their PS complexes are very effective in capturing light energy due to the large carotenoid antenna and efficient energy transfer among the LH1 antenna. The optical cross-section

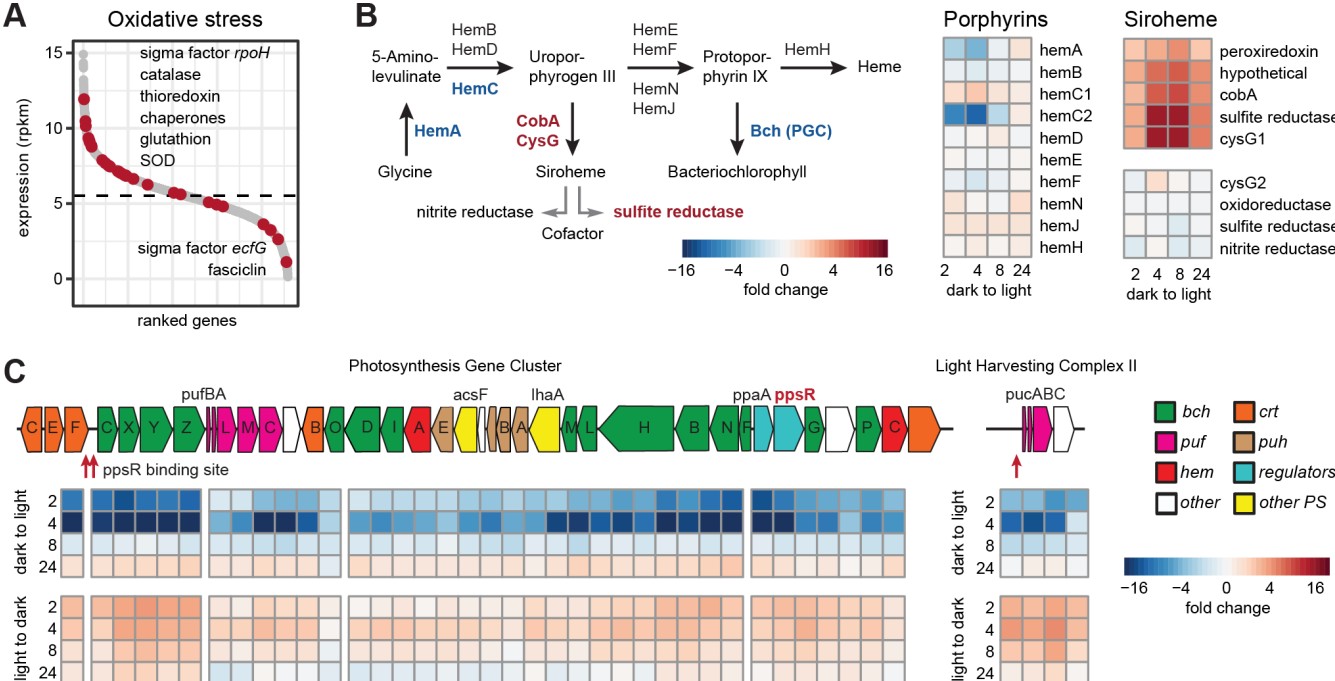

**FIG 6** Response of stress response, porphyrin biosynthesis pathway, and PGC to changing light regimes. (A) Average expression of all (gray) and oxidative stress response genes (red) as read counts per kilobase gene length and million counts (rpkm). (B) Panel on the left depicts the common porphyrin biosynthesis pathway. Up- and downregulated genes are colored in red and blue, respectively. The panel on the right shows expression changes of the involved genes and operons after the transition to light. (C) Expression dynamics in both time series for the PGC and *puc* operon. Genes are colored based on their function (*bch*, Bchl biosynthesis; *crt*, carotenoid biosynthesis; *puf*, *puh*, structural proteins; *hem*, heme biosynthesis; and *regulators*, *ppsR*, *ppaA*, and *tspO*). Putative binding sites for PpsR are marked in red.

of the RC-LH1 complex is even larger than the one measured in *Gemmatimonas phototrophica* assembling a double ring of light-harvesting antenna encircling the RC (31) and more than four times larger than in *Rhodospirillum rubrum* (32). The high and continuous abundance of RC-LH1 with a large effective antenna size, therefore, makes KRV36 exceptionally efficient at collecting the almost continuous light of moderate intensity that illuminates its natural habitat. In addition, KRV36 cells also contain large amounts of carotenoids, which are not coupled to photosynthetic complexes, and, therefore, might have a photo-protective rather than light-harvesting function.

Another interesting feature is the presence of ICM vesicles (chromatophores), which are common features in many purple non-sulfur bacteria (33) but, as yet, have never been convincingly demonstrated in AAP bacteria. The chromatophores are preferentially located at both poles of the cells throughout their whole cell cycle. Such bipolar arrangement of PS membranes is rare even among purple non-sulfur bacteria and has so far only been observed for lamellae ICM but not spherical chromatophores (34). Thus, KRV36 shares some characteristics with AAP bacteria (reduced pigment content compared to the purple non-sulfur bacteria, high amount of carotenoids), but with the abundant presence of chromatophores, the newly described arctic phototrophic strain resembles more purple non-sulfur bacteria than common AAP species.

Aerobically grown KRV36 cells were pigmented and photosynthetically active even when grown under continuous light. This sharply contrasts with the AAP bacteria described so far, in which the transcription of photosynthesis genes is repressed and cells lose their pigmentation completely after a few hours of illumination (15–20). Such a regulatory system would effectively prevent any pigment synthesis during arctic summer with almost no dark period. As a result, the common AAP bacterium would be able to assemble and use its PS apparatus only during days when solar light would be sufficiently reduced due to cloud cover. Naturally, such environmental conditions will

impose selective pressure on species that are capable of continuous pigment synthesis. There are two possible, non-exclusive explanations of how the ability to produce BChl in light evolved in our strain. In contrast to other AAP bacteria studied so far (17, 35), the regulatory genes *ppsR* and *ppaA* are strongly repressed upon illumination. Assuming that the high transcription of both genes is balanced by a high proteolysis rate, the actual protein concentration would diminish over time resulting in a de-repression of the target genes. The number of PpsR-binding motifs in the PGC and LHII promoters (three) is lower compared to *Dinoroseobacter shibae* (eight), *Rhodobacter* sphaeroides (seven), and *Rhodopseudomonas palustris* (seven) as representatives of aerobic and anaerobic anoxygenic phototrophs, respectively (19, 25, 36). This could further contribute to the observed diminishing of the repressive effect. It also needs to be clarified if the ability to permanently repress the PGC is completely lost or if only the light sensitivity of the regulatory system is attenuated.

Conducting BChl synthesis in the presence of oxygen and light poses the risk of ROS formation by BChl precursors. A high oxygen concentration may increase the risk that the normal photo-protective function of carotenoids in the PS complexes, to quench triplet BChl before the ROS singlet oxygen can form, may be overwhelmed. Interestingly, the singlet-oxygen specific regulatory system common in anoxygenic phototrophic *Rhodobacteraceae*, sigma factor RpoE, and its repressor ChrR (14, 28, 37) is absent in KRV36. The presence of a large amount of free carotenoids in the PS membranes, which has also been observed in other AAP bacteria (17, 38), may also have a photo-protective function by directly scavenging any singlet oxygen or other ROS species (39, 40). A constitutively high expression of the general oxidative stress response under the control of sigma factor RpoH (41) and with the enzymes catalase and superoxide dismutase as well as the thioredoxin (42) and glutaredoxin (43) systems at its core might keep the cells permanently adapted to prevent the detrimental effects of ROS formation in the respiratory chain components and elsewhere in the cell. With a full set of responsive genes conserved in bacteria (44) active, KRV36 can cope with light exposure without being apparently stressed. Rerouting of porphyrin biosynthesis through fast transcriptional activation of cobalamin and heme pathways is controlled by RpoE in *D. shibae* (19). In KRV36, transient activation of siroheme synthesis in the light is inverse to the BChl repression curve. Thus, a simple balancing mechanism might be in place, which is not directly controlled by the stress response system. Small shifts in the ratios of enzymes at branching points could help to fine-tune the substrate flow through the pathway according to slightly changing needs.

The tendency of KRV36 cells to form cell clumps is also notable. The environment within cell aggregates is anoxic. Furthermore, we isolated our strain from a cyanobacterial biofilm. Its natural habitat in aggregation with other cells might reduce light and oxidative stress to an easily bearable level. At the moment, we can only speculate if the spikes observed at the surface of the cells play any role in cell adhesion and whether there is any active regulation of this process. In *Azospirillum brasilense,* flocculation, similar to that observed in KRV36, is mediated by fasciclin under the transcriptional control of the RpoE-ChrR system (45). However, the fasciclin-coding gene of KRV36 was not expressed in the analyzed transcriptomes. It might either be not involved in the observed cell aggregation or expression occurred outside our sampling frame.

In conclusion, the presented study documents a remarkable example of the plasticity of a phototrophic species to adapt its gene regulatory mechanism to overcome challenging environmental conditions (summarized in Fig. 7). We argue that the specific traits observed in *Sediminicoccus* sp. KRV36, large absorption cross-section, ability to synthesize its photosynthetic apparatus under light, the presence of protective carotenoids, and robust oxidative stress protection, allow this organism to thrive and utilize photosynthesis under challenging conditions of the polar summer.

## MATERIALS AND METHODS

### Sampling site and strain isolation

Samples were collected in northern Iceland in the littoral of a right tributary of the Kollavíkurá stream above the valley of lake Kollavíkurvatn (66°16′22.4″N, 15°48′24.5″W) in July 2020. Samples containing mats of nostocacean cyanobacteria were homogenized by gentle pipetting and serially diluted ($10^{-1}$-$10^{-5}$) for inoculation onto half-strength Reasoner's 2A medium agar. BChl-containing colonies detected by an infra-red fluorescence imaging system (46) were further purified by passaging to receive single-cell colonies. The identity of the KRV36 strain was revealed by amplification and sequencing of its 16S rRNA gene.

### Culture growth and sampling

KRV36 was grown at 23°C in half-strength R2A medium supplemented with 160 mg of $MgCl_2$, 60 mg of $CaCl_2$, 10 mL of SL-4 micronutrients (cf. DSMZ medium 462), and 2 mL of vitamins (cf. DSMZ medium 462) per liter of full medium. Batch cultures continuously illuminated by LED panels providing 100 µmol photon $m^{-2}$ $s^{-1}$ of white light were aerated by bubbling and stirring (<100 rpm).

#### Effect of light and dark on photosynthesis gene expression.

Erlenmeyer flasks containing 100 mL of medium were inoculated with 2.5 mL of a pre-culture and placed on an orbital shaker (150 rpm). Two flasks were incubated in the dark, two were exposed to light intensities of 100 µmol photon $m^{-2}$ $s^{-1}$ for 72 hours. Then, the samples were collected from all four flasks, the cultures' light regime was reversed, and the sampling followed the cultures after 2, 4, 8, 12, 16, 20, and 24 hours into their dark-light and light-dark transitions, respectively.

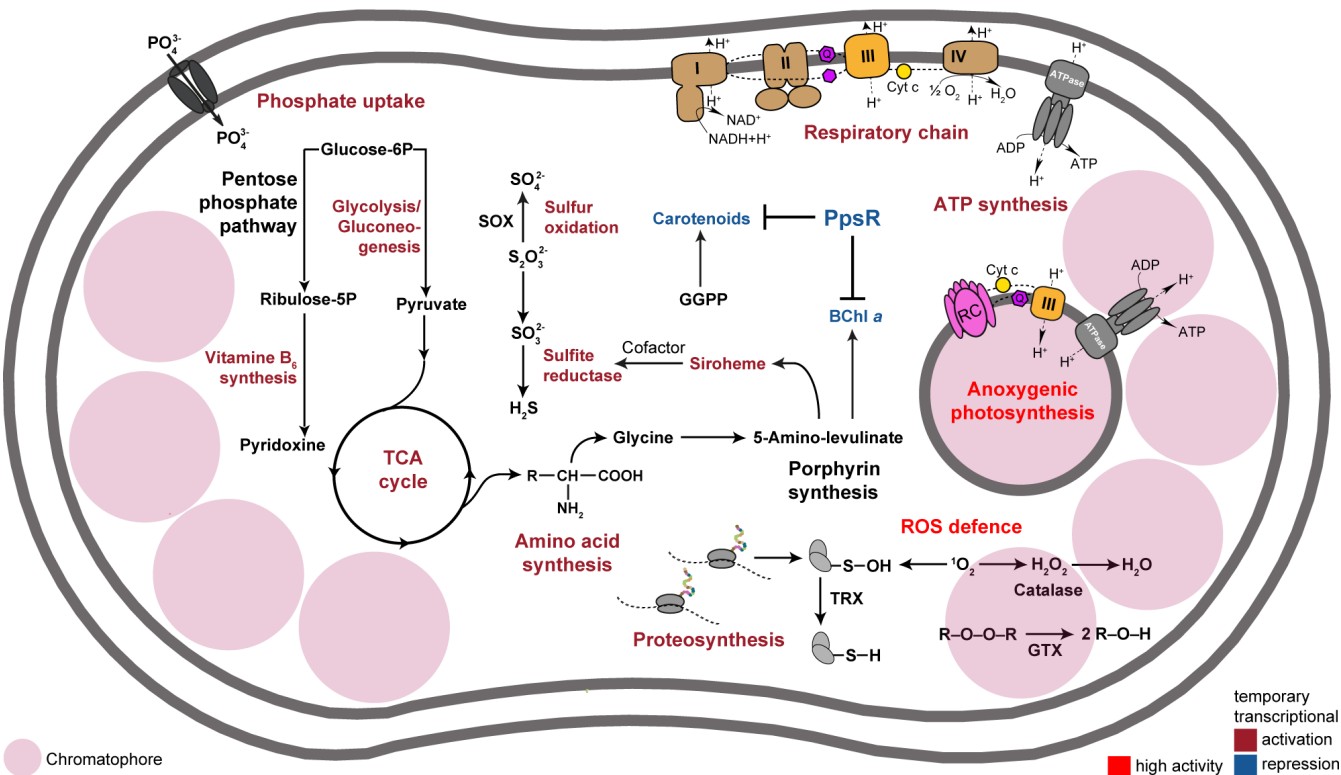

**FIG 7** Response of selected metabolic pathways after transition to light. Increased and decreased gene expression of pathways is indicated by red and blue, respectively. High constitutive activity is indicated in bright red. Cyt, terminal cytochrome oxidase; GGPP, geranylgeranyl diphosphate; GTX, glutathione; Q, quinone pool; RC, reaction center; SOX, sulfur/thiosulfate oxidation system; TCA cycle, citrate cycle; and TRX, thioredoxin.

## Genome sequencing

### DNA extraction

Genomic DNA was extracted and purified using the TIANamp Genomic DNA Kit (TIANGEN Biotech, Beijing, China). High molecular weight genomic DNA was extracted as previously described (47). The identity of the strain was verified by amplification and sequencing of its 16S rRNA gene. The complete genome of KRV36 was sequenced by combining Oxford Nanopore and Illumina NovaSeq 6000. The obtained reads were quality filtered with Trimmomatic (48) and assembled using Unicycler (49) in default mode. Annotation was carried out using the NCBI Prokaryotic Genome Annotation Pipeline (version 6.1, https://www.ncbi.nlm.nih.gov/genome/annotation_prok/). For metabolic pathway predictions, we combined the automatic annotations provided by the Kyoto Encyclopedia of Genes and Genomes (50) with manual curation and searches for missing genes using BLAST (51).

## Phylogenetic analysis

The 16S rRNA gene sequences of KRV36 (GenBank accession number: LHU95_16785) and *S. rosea* R-30[T] (GenBank accession number: R9Z33_03575) were retrieved from their genome sequences (GenBank accession numbers: CP085081.1 and CP137852-CP137853, respectively). Reference sequences were obtained either from the SILVA database (52) or NCBI GenBank (July 2021). Nucleotide sequences were aligned using ClustalW, and the phylogenetic tree was calculated using both neighbor-joining (53) and maximum likelihood (54) algorithms included in the MEGA 6.06 software (55). Bootstrap resampling for 1,500 replicates was performed. *Roseomonas gilardii* NBRC 107871[T] was used as an outgroup. The calculation of pairwise 16S rRNA gene sequence similarity between KRV36 and its closest relative *S. rosea* R-30[T] was done in the EzTaxon server (56).

## Microscopy

### Epifluorescence microscopy

BChl and DAPI fluorescence was visualized using an Axio Zeiss Imager.Z2 microscope (Carl Zeiss AG, Jena, Germany) equipped with alpha Plan-Apochromat 63×/1.46 oil objective and an EM-CCD digital camera C9100 (Hamamatsu, Japan). DAPI emission was recorded at 420–470 nm, while BChl autofluorescence was detected above 850 nm.

### Array tomography

Cell pellets were fixed in 2.5% glutaraldehyde in phosphate buffer, post-fixed in 2% $OsO_4$ for 2 hours, dehydrated in graded series of acetone, embedded, and infiltrated in SPURR resin. Seventy nanometer ultrathin sections were cut by ultramicrotome Leica UTC-6 (Danaher Corp., Washington, DC, USA) using a specialized diamond knife (57), collected on a negatively charged silicon wafer, and post-stained with uranyl acetate for 30 min and carbon coated. Array sections were imaged using Apreo scanning electron microscope and MAPS software (Thermo Fisher Scientific, Waltham, MA, USA). Serial images were acquired using the optiplan mode and the following parameters: accelerating voltage 2.5 keV, probe current 0.4 nA, WD 3 mm, resolution 4 nm, stage bias −4,000 V, compound lens filter 1.4 keV, slice thickness 70 nm, and dwell time per pixel 6 µs. Tomogram image data were processed by TrakEM2 (58), Microscopy Image Browser (59), and Amira. Sub-volumes of selected bacteria were cut out from the tomogram, semi-automatically aligned along the *Z*-axis, followed by segmentation, 3D model generation, and volume analysis, all using the IMOD (60). Surface structures automatically filtered by Noise Despeckle, Threshold (Renyi Entropy), Erode, Dilate, Fill Holes, and Invert contrast were measured using ImageJ (61).

## Cryo-electron microscopy

PS complexes were isolated and purified as described earlier (31). To obtain a contour density map, the purified PS complex of the reaction center (RC) and light-harvesting 1 (LH1) antenna (the complex is termed RC-LH1) was deposited on glow-discharged holey carbon grids and rapidly frozen in liquid ethane with plunge freezer LEICA EM GP2 (Danaher Corp., Washington, DC, USA). For sample visualization, a 200 kV JEOL JEM-2100F TEM was used (JEOL, Tokyo, Japan) at 25,000 × magnification. TEM images were recorded using a Gatan K2 Summit direct detection camera, with a resolution corresponding to 1.4 Å per pixel. A data set of 200,000 particles was collected, and image analysis was carried out using cryoSPARC.

## Atomic force microscopy

Cells were immobilized onto clean glass slides using CellTak (Merck, Germany) and imaged in NanoWizard 4Bio AFM (Bruker, USA) using Quantitative Imaging mode under ambient temperature and buffered with 150 mM KCl and 10 mM Tris.Cl pH 7.5. PFQNM-LC-A-CAL cantilevers (Bruker, USA) with tip length of 17 µm were calibrated using the method by Sader et al. (62) yielding a spring constant $k = 0.033$ N·m$^{-1}$ and a resonant frequency $f_0 = 39.8$ kHz in air and $f_0 = 20.1$ kHz in the buffer. The $6 \times 6$ µm$^2$ topography images were scanned with a resolution of $256 \times 256$ pixels$^2$ at a setpoint of 80 pN and the tip vertical speed was 83.3 µm s$^{-1}$.

## *In vivo* measurements

### BChl fluorescence

Kinetic fluorometer FL-3000 (Photon Systems Instruments Ltd., Czech Republic) was used as described in Kaftan et al. (63). Cells harvested in the late exponential phase were resuspended in fresh medium and dark adapted for half an hour at room temperature under aerobic conditions. Kinetics of the fast fluorescence induction was elicited by single-turnover saturating flash (50 µs, 0.33 mol photon m$^{-2}$ s$^{-1}$). BChl fluorescence signal ($\lambda > 850$ nm) was detected with a 10 MHz temporal resolution.

### Oxygen respiration

Four milliliters of cells resuspended in fresh medium was placed in the OX1LP chamber (Qubit Systems Inc., Kingston, Canada) at 25°C. The measurements were conducted in the dark and under increasing intensities of white light (50–1,000 µmol photon m$^{-2}$ s$^{-1}$) provided by an A113 LED light source (Qubit Systems Inc., Kingston, Canada).

### Oxygen concentration in aggregates

Measurements of oxygen concentration inside the cell aggregates (2 mm wide disc shape of 1 mm height) placed on top of 8 mm diameter oxygen-sensing luminescence patches using Neofox-Kit-Probe optrode system (Ocean Insight, USA) showed no traces of oxygen at the center of the aggregate.

## Reverse transcription quantitative PCR

The RevertAid First Strand cDNA Synthesis Kit (Thermo Fisher Scientific, USA) was used for cDNA generation from 200 ng of RNA with pre-incubation of the RNA template and random hexamers for 10 min at 65°C. Relative quantification of *pufM* and *ppsR* transcripts was performed in triplicates in a CFX Connect Real-Time PCR cycler (Bio-Rad Laboratories, Inc., Hercules, CA, USA) using the PowerUp Sybr Green kit (Applied Biosystems, USA). The *rpoD* gene was used as a reference. The comparative $C_t$ method (64) was used to quantify changes in gene expression. Specific primers and PCR conditions are listed in Table S1.

## RNA sequencing

Cells from triplicate cultures were harvested by centrifugation, resuspended in 1 mL of PGTX extraction solution (65), and immediately frozen in liquid nitrogen. RNA was extracted as described earlier (66). Libraries were generated according to the protocol of Shishkin et al. (67), including rRNA removal with the RiboZero Kit (Illumina, USA) and sequenced on a NovaSeq 6000 (Illumina, USA) in paired-end mode with 100 cycles in total. Raw reads were processed, and differential gene expression was assessed as described before (66). Briefly, quality-filtered reads were mapped to the KRV36 genome (GenBank accession: CP085081.1) using bowtie2. FeatureCounts was used to assess the number of reads per gene. Normalization and identification of significantly differentially regulated genes (false discovery rate, FDR < 0.05) were performed with edgeR. For the analysis of significantly enriched metabolic pathways, the corresponding data were downloaded from the KEGG database (https://www.kegg.jp), and the hypergeometric test including correction for FDR was employed in R. Binding sites for PpsR were obtained from the Prodoric database (68).

## ACKNOWLEDGMENTS

This research was supported by Czech-BioImaging project of the Ministry of Education (LM2018129 Czech-BioImaging) (D.K. and T.B.), European Regional Development Fund-Project No. CZ.02.1.01/0.0/0.0/15_003/0000441 (D.K., T.B., and Z.G), International Network for Terrestrial Research and Monitoring in the Arctic (INTERACT III-EU H2020) Transnational Access-AETHER (DK), and CSF PhotoGemm +project GX19-28778X (J.T., K.K., A.T.G., N.S., and M.K.).

We acknowledge the BC CAS core facility LEM supported by Czech-Bioimaging project No. LM2023050 and European Regional Development Fund-Project No. CZ.02.1.01/0.0/0.0/18_046/0016045. The authors thank Jason Lawrence Dean and Otakar Strunecký for conducting light microscopy and Jiří Týč for recording SEM tomography.

## AUTHOR AFFILIATIONS

[1]Laboratory of Anoxygenic Phototrophs, Institute of Microbiology of the Czech Academy of Sciences, Třeboň, Czechia

[2]Institute of Parasitology, Biology Centre, Czech Academy of Sciences, České Budějovice, Czechia

[3]Department Chemistry, Faculty of Science, University of South Bohemia, České Budějovice, Czechia

## AUTHOR ORCIDs

Karel Kopejtka http://orcid.org/0000-0001-5412-4541
Tomáš Bílý http://orcid.org/0000-0003-1450-1693
Michal Koblížek http://orcid.org/0000-0001-6938-2340
David Kaftan http://orcid.org/0000-0003-0932-0986

## FUNDING

| Funder | Grant(s) | Author(s) |
| --- | --- | --- |
| Ministerstvo Školství, Mládeže a Tělovýchovy (MŠMT) | LM2018129 | Tomáš Bílý |
| | | David Kaftan |
| Ministerstvo Školství, Mládeže a Tělovýchovy (MŠMT) | CZ.02.1.01/0.0/0.0/15_003/0000441 | Tomáš Bílý |
| | | Zdenko Gardian |
| | | David Kaftan |
| Grantová Agentura České Republiky (GAČR) | GX19-28778X | Michal Koblížek |

| Funder | Grant(s) | Author(s) |
|---|---|---|
| Ministerstvo Školství, Mládeže a Tělovýchovy (MŠMT) | LM2023050 | Zdenko Gardian |
| Ministerstvo Školství, Mládeže a Tělovýchovy (MŠMT) | CZ.02.1.01/0.0/0.0/18_046/0016045 | Zdenko Gardian |

## AUTHOR CONTRIBUTIONS

Jürgen Tomasch, Conceptualization, Data curation, Formal analysis, Investigation, Methodology, Software, Validation, Visualization, Writing – original draft, Writing – review and editing | Karel Kopejtka, Data curation, Formal analysis, Investigation, Methodology, Validation, Visualization, Writing – original draft, Writing – review and editing | Tomáš Bílý, Data curation, Formal analysis, Investigation, Methodology, Validation, Visualization, Writing – review and editing | Alastair T. Gardiner, Data curation, Formal analysis, Investigation, Methodology, Validation, Visualization, Writing – original draft, Writing – review and editing | Zdenko Gardian, Data curation, Formal analysis, Visualization, Writing – review and editing | Sahana Shivaramu, Data curation, Formal analysis, Investigation, Validation, Visualization | Michal Koblížek, Conceptualization, Data curation, Formal analysis, Funding acquisition, Project administration, Resources, Supervision, Validation, Visualization, Writing – original draft, Writing – review and editing | David Kaftan, Conceptualization, Data curation, Formal analysis, Funding acquisition, Investigation, Methodology, Resources, Validation, Writing – original draft, Writing – review and editing

## DATA AVAILABILITY

The complete genome sequences of the KRV36 and R-30 strains are deposited at NCBI GenBank under the accession numbers CP085081.1 and CP137852-CP137853, respectively. RNA sequencing data are publicly available at the NCBI gene expression omnibus database under accession number GSE245756.

## ADDITIONAL FILES

The following material is available online.

### Supplemental Material

**Supplemental material (mSystems01311-23-s0001.docx).** Tables S1 to S3, Figures S1 to S7, and Text S1.
**Table S4 (mSystems01311-23-s0002.xls).** Metabolic pathways in the *S. rosea* KRV36 genome.
**Table S5 (mSystems01311-23-s0003.xlsx).** Transcriptome of *S. rosea* KRV36.

### Open Peer Review

**PEER REVIEW HISTORY (review-history.pdf).** An accounting of the reviewer comments and feedback.

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
