## [Reviewer comments · mSystems]

A photoheterotrophic bacterium from Iceland has adapted its photosynthetic machinery to the long days of polar summer

Jürgen Tomasch, Karel Kopejtko, Tomáš Bílý, Alastair Gardiner, Zdeno Gardian, Sahana Shivaramu, Michal Koblizek, and David Kaftan

Corresponding Author(s): David Kaftan, Mikrobiologický ústav Akademie věd České republiky

Review Timeline:

Submission Date:	December 5, 2023
Editorial Decision:	January 9, 2024
Revision Received:	January 16, 2024
Accepted:	January 26, 2024

Editor: Angela Re

Reviewer(s): Disclosure of reviewer identity is with reference to reviewer comments included in decision letter(s). The following individuals involved in review of your submission have agreed to reveal their identity: Xiaoling Xu (Reviewer #2); Joval Navas Martinez (Reviewer #3)

Transaction Report:

DOI: <https://doi.org/10.1128/msystems.01311-23>

Re: mSystems01311-23 (A photoheterotrophic bacterium from Iceland adapted its photosynthetic machinery to long days of polar summer)

Dear Dr. David Kaftan:

Revision Guidelines

Sincerely,
Angela Re
Editor
mSystems

Reviewer #1 (Comments for the Author):

In this article, Tomasch et al. isolated a BChl-producing bacterium KRV36 from an oligotrophic stream near Arctic Circle. The structure and morphology of the cell, especially, the photosynthetic complexes and photosystem activities were revealed to explore the strain's adaptation to the polar light conditions. Both the writing and the protocol in this manuscript seem to be impeccable, almost perfect. However, in the "Discussion" section, I see a little more conclusions or explanations from other people's reports than your own data analysis.

So, it might be better to explain the possible reasons why pufM expression was lowered significantly in the point of two hours after the dark-light transition, however, with no significantly differentially expressed genes in initial transcriptome. What's a possible mechanism of the transient cellular photodamage repair? And, the relationship with the adaptation of prolonged light exposure? What a possible roles of oxidative stress response proteins and other peripheral light harvesting proteins in this process?

It may be even more appealing, if a possible photodamage repair process or mechanism could be described or assumed in more detail.

Reviewer #2 (Comments for the Author):

In this manuscript, Tomasch et al. describe a new photoheterotrophic *Sediminicoccus* sp. strain KRV36, which was isolated from a cold stream located near Raufarhöfn in north-western Iceland, an area characterized by a long summer light period and almost no darkness. In contrast to most aerobic anoxygenic phototrophs (AAPs), it expresses photosynthesis genes, synthesizes bacteriochlorophyll and assembles functional photosynthetic complexes under continuous light in the presence of oxygen. To understand the strain's adaptation to the polar light conditions, the authors analyzed the structure and function of the photosystem, sequenced the genome, and characterized the transcriptional correlation between the ROS-related genes and photosystem gene clusters (PGCs) to changes in illumination.

Overall, the topic of this study is interesting, it is of great interest to readers in the field of prokaryotic photosynthesis. The results are informative and well-supported, and will broaden our understanding of the adaptation strategies of the arctic phototrophs in response to extreme environmental conditions.

However, some of the conclusions are overstated, and need to be clarified and tuned down:

1. Line 257: The author stated that KRV36 rapidly synthesized siroheme for a short period of light exposure, rather than synthesizing BChl and heme, to resist ROS. However, this conclusion is derived from the transcription levels of the ROS-related genes, photosystem gene clusters (PGCs), and the heme-synthesizing genes. These analyses are not sufficient to support the statement. The authors are recommended to tune down the statement or to further clarify the correlations between ROS and PGC.

2. The authors are recommended to pay attention to the discussion. Some of the statements need to be tuned down. For example:

(1) Line 332: "In KRV36, transient activation of siroheme synthesis in the light is inverse to the BChl repression curve. Thus, a simple balancing mechanism might be in place that is not directly controlled by the stress response system."

(2) Line 335: The formation of cell aggregates in lab culturing does not represent the natural state of the cells. Therefore, the statement concerning cell aggregates as an additional protective mechanism against light stress using self-shading needs to be carefully considered and revised.

3. Line 243: "Transcriptional activation of oxidative stress response genes was not detected upon light exposure." Are the transcription levels of oxidative stress response genes changed during light to dark transitions?

4. Line 268: Regarding the number of PpsR binding sites to PGC-related operons, has this been compared to other AAP species or purple non-sulfur bacteria? The authors mentioned in the Discussion section (lines 311-312) that the PpsR binding motifs in the PGC promoters are less conserved, is this correlated to the ability of PpsR in regulating the gene expression of photosynthetic pigments?

5. Fig. 1D is not cited in the manuscript and should be added in Line 121.

6. Figure 5B: The authors are recommended to specify the name of the four gene clusters (cluster1/2/3/4).

7. Fig6C: bch, crt, puf, puh, etc. should be labeled with the corresponding gene name in the legend.

8. Figure S6B: The figure legend needs to be clarified. Do the blue-green and orange-red lines represent the two biological replicates?

Reviewer #3 (Comments for the Author):

The submitted manuscript presents the unique characteristics of the photoheterotrophic bacterium *Sediminicoccus* sp. strain KRV36, isolated from a cold stream in Iceland. It highlights its remarkable ability to maintain photosynthesis gene expression, synthesize bacteriochlorophyll, and assemble functional photosynthetic complexes even under continuous light in the presence of oxygen, which sets it apart from other Aerobic Anoxygenic Phototrophs.

The data presented has well established the current objective of the study, which is to understand the strain's adaptation to the polar light conditions. A few questions might be helpful to be clarified in the paper:

1. Please clarify line 337... it may lower the oxygen concentration in the aggregates... Have investigations been conducted to measure oxygen concentration before and after aggregation?

2. Regarding line 338, the speculation regarding the potential role of spikes in cell adhesion is intriguing. Are there identified candidate genes within KRV36 potentially associated with this speculated mechanism of action?

Editor (Comments for the Author):

The authors are invited to detail further their description of the identification of the strain KRV36 and of the metabolic pathway analysis in the methodological section.

In this article, Tomasch et al. isolated a BChl-producing bacterium KRV36 from an oligotrophic stream near Arctic Circle. The structure and morphology of the cell, especially, the photosynthetic complexes and photosystem activities were revealed to explore the strain's adaptation to the polar light conditions. Both the writing and the protocol in this manuscript seem to be impeccable, almost perfect. However, in the "Discussion" section, I see a little more conclusions or explanations from other people's reports than your own data analysis.

So, it might be better to explain the possible reasons why pufM expression was lowered significantly in the point of two hours after the dark-light transition, however, with no significantly differentially expressed genes in initial transcriptome. What's a possible mechanism of the transient cellular photodamage repair? And, the relationship with the adaption of prolonged light exposure? What a possible roles of oxidative stress response proteins and other peripheral light harvesting proteins in this process?

It may be even more appealing, if a possible photodamage repair process or mechanism could be described or assumed in more detail.

Reviewer #1 (Comments for the Author):

In this article, Tomasch et al. isolated a BChl-producing bacterium KRV36 from an oligotrophic stream near Arctic Circle. The structure and morphology of the cell, especially, the photosynthetic complexes and photosystem activities were revealed to explore the strain's adaptation to the polar light conditions. Both the writing and the protocol in this manuscript seem to be impeccable, almost perfect. However, in the "Discussion" section, I see a little more conclusions or explanations from other people's reports than your own data analysis.

Thank you for the positive evaluation of our manuscript. We added references on the role of carotenoids in ROS defense, the distribution of defense systems in bacteria and on the proposed regulation by ppsR in another representative anoxygenic phototroph.

So, it might be better to explain the possible reasons why pufM expression was lowered significantly in the point of two hours after the dark-light transition, however, with no significantly differentially expressed genes in initial transcriptome.

response R#1-1: We now explain better our analysis that aimed at identifying highly variable gene expression as outlined in Supplementary Figure S6 in the respective figure legend. We assessed significance of gene expression changes only for the first two time-points with an overlapping time course for both replicates. For determining expression at the later time points we kept in mind the differences in the two time-courses for the identified highly variable genes (in particular the PGC) and further assessed transcription using RT-qPCR.

What's a possible mechanism of the transient cellular photodamage repair? And, the relationship with the adaption of prolonged light exposure? What a possible roles of oxidative stress response proteins and other peripheral light harvesting proteins in this process?

response R#1-2: Thank you for these questions. We reason that a constitutively active general stress response system, detoxifying the cell and repairing damaged cellular components using glutathione and thioredoxin helps to deal with the generated ROS, also beyond singlet oxygen. We clarified now that the stress response genes showed a high expression at all time-points (line 243). Furthermore, we think that the high carotenoid content might be involved in scavenging ROS. We did not support this claim by literature before. We now cite two according studies in the revised manuscript.

It may be even more appealing, if a possible photodamage repair process or mechanism could be described or assumed in more detail.

response R#1-3: We agree that it will be highly interesting to unravel the oxidative stress response of our *Sediminicoccus* strain in greater detail. We therefore plan to study the physiological and regulatory effects of different types of oxidative stress in future. For the current manuscript we

would like to restrict the analysis to a description of the most abundant systems based on gene expression.

Reviewer #2 (Comments for the Author):

In this manuscript, Tomasch et al. describe a new photoheterotrophic *Sediminicoccus* sp. strain KRV36, which was isolated from a cold stream located near Raufarhöfn in north-western Iceland, an area characterized by a long summer light period and almost no darkness. In contrast to most aerobic anoxygenic phototrophs (AAPs), it expresses photosynthesis genes, synthesizes bacteriochlorophyll and assembles functional photosynthetic complexes under continuous light in the presence of oxygen. To understand the strain's adaptation to the polar light conditions, the authors analyzed the structure and function of the photosystem, sequenced the genome, and characterized the transcriptional correlation between the ROS-related genes and photosystem gene clusters (PGCs) to changes in illumination.

Overall, the topic of this study is interesting, it is of great interest to readers in the field of prokaryotic photosynthesis. The results are informative and well-supported, and will broaden our understanding of the adaptation strategies of the arctic phototrophs in response to extreme environmental conditions.

Thank you. We appreciate the critical and constructive evaluation of our manuscript.

However, some of the conclusions are overstated, and need to be clarified and tuned down:

1. Line 257: The author stated that KRV36 rapidly synthesized siroheme for a short period of light exposure, rather than synthesizing BChl and heme, to resist ROS. However, this conclusion is derived from the transcription levels of the ROS-related genes, photosystem gene clusters (PGCs), and the heme-synthesizing genes. These analyses are not sufficient to support the statement. The authors are recommended to tune down the statement or to further clarify the correlations between ROS and PGC.

response R#2-1: We are sorry that the choice of sectioning our results led to a misunderstanding. Here, we intended only to describe the expression of the different systems without drawing further conclusions on possible connections. We created now three separate subsections. In the discussion we actually come to the conclusion that the observed expression pattern of porphyrin metabolism genes is not related to oxidative stress, in contrast to *D. shibae* where a clear connection can be found. We agree that transcriptional data alone does not allow to draw broad conclusions on the actual biosynthesis pathway activity (see next point).

2. The authors are recommended to pay attention to the discussion. Some of the statements need to be tuned down. For example:

(1) Line 332: "In KRV36, transient activation of siroheme synthesis in the light is inverse to the BChl repression curve. Thus, a simple balancing mechanism might be in place that is not directly controlled by the stress response system."

response R#2-2(1): The expression changes of the branching node genes were actually quite strong and we think it is reasonable to propose a balancing mechanism here. However, we consider

now that the actual changes in transcription might not lead to drastic changes in enzyme concentration or substrate turnover. The discussion now reads: "Rerouting of porphyrin biosynthesis through fast transcriptional activation of cobalamin and heme pathways is controlled by RpoE in *D. shibae* (Tomasch et al., 2011). In KRV36, transient activation of siroheme synthesis in the light is inverse to the BChl repression curve. Thus, a simple balancing mechanism might be in place that is not directly controlled by the stress response system. Small shifts in the ratios of enzymes at branching points could help to fine-tune the substrate flow through the pathway according to slightly changing needs."

(2) Line 335: The formation of cell aggregates in lab culturing does not represent the natural state of the cells. Therefore, the statement concerning cell aggregates as an additional protective mechanism against light stress using self-shading needs to be carefully considered and revised.

response R#2-2(2): We agree on this point and removed the statement on self-shading. We now discuss the point of aggregation from a different angle. As suggested by another reviewer we showed in an additional experiment that the environment within cell colonies is anoxic. We also need to consider the isolation site. The discussion now reads: "The tendency of KRV36 cells to form cell clumps is also notable. The environment within cell aggregates is anoxic. Furthermore, we isolated our strain from a cyanobacterial biofilm. Its natural habitat in aggregation with other cells might reduce light and oxidative stress to an easily bearable level."

3. Line 243: "Transcriptional activation of oxidative stress response genes was not detected upon light exposure." Are the transcription levels of oxidative stress response genes changed during light to dark transitions?

response R#2-3: No, the oxidative stress response genes are stably expressed in all samples. We refined the sentence: "Despite the expected higher risk of oxidative stress upon light exposure, we saw no transcriptional changes of the oxidative stress response genes under changing illumination."

4. Line 268: Regarding the number of PpsR binding sites to PGC-related operons, has this been compared to other AAP species or purple non-sulfur bacteria? The authors mentioned in the Discussion section (lines 311-312) that the PpsR binding motifs in the PGC promoters are less conserved, is this correlated to the ability of PpsR in regulating the gene expression of photosynthetic pigments?

response R#2-4: Thank you for this question. To our knowledge no comprehensive analysis of ppsR binding sites has been published so far. We used *Dinoroseobacter shibae*, *Rhodobacter sphaeroides* and *Rhodospirillum rubrum* as models because they are well studied representatives; transcriptome data are available also for the first two cases. We added the numbers of the binding sites for these strains that we determined using Prodigal search, to the discussion. A broader analysis of binding site evolution should be clearly addressed in the future.

5. Fig. 1D is not cited in the manuscript and should be added in Line 121.

response R#2-5: Thank you for noticing. The citation of the Fig 1D was added to the line 121

6. Figure 5B: The authors are recommended to specify the name of the four gene clusters (cluster1/2/3/4).

response R#2-6: We refer to the genes within one of the four highlighted clusters in the text (lines 223 ff). Each cluster consists of genes with similar expression but not necessarily from the same location and with a specified cluster name. Cluster 4 in the Figure is the PGC, but also the LHII operon.

7. Fig6C: bch, crt, puf, puh, etc. should be labeled with the corresponding gene name in the legend.

response R#2-7: the abbreviations used for the color-coding of gene types are now detailed in the figure legend.

8. Figure S6B: The figure legend needs to be clarified. Do the blue-green and orange-red lines represent the two biological replicates?

response R#2-8: We now specified in the legend that the data show two biological replicate time series for each transition. As suggest by another reviewer we also added text clarifying our choice for statistical analysis.

Reviewer #3 (Comments for the Author):

The submitted manuscript presents the unique characteristics of the photoheterotrophic bacterium *Sediminicoccus* sp. strain KRV36, isolated from a cold stream in Iceland. It highlights its remarkable ability to maintain photosynthesis gene expression, synthesize bacteriochlorophyll, and assemble functional photosynthetic complexes even under continuous light in the presence of oxygen, which sets it apart from other Aerobic Anoxygenic Phototrophs.

The data presented has well established the current objective of the study, which is to understand the strain's adaptation to the polar light conditions.

Thank you for the positive evaluation of our manuscript.

A few questions might be helpful to be clarified in the paper:

1. Please clarify line 337... it may lower the oxygen concentration in the aggregates... Have investigations been conducted to measure oxygen concentration before and after aggregation?

response R#3-1: we now carried out measurements of oxygen concentration inside the cell aggregates (2 mm wide disc shape of 1 mm height) placed on top of 8 mm diameter oxygen sensing luminescence patches using Neofox-kit-probe optrode system (Ocean Insight, USA) that showed no traces of oxygen at the center of the aggregate (Methods). We consider this information in the discussion: "The tendency of KRV36 cells to form cell clumps is also notable. The environment within cell aggregates is anoxic. Furthermore, we isolated our strain from a cyanobacterial biofilm. Its natural habitat in aggregation with other cells might reduce light and oxidative stress to an easily bearable level."

2. Regarding line 338, the speculation regarding the potential role of spikes in cell adhesion is intriguing. Are there identified candidate genes within KRV36 potentially associated with this speculated mechanism of action?

response R#3-2: At this moment we could only speculate about certain candidates. The fasciclin gene with a proven role in light-induced aggregation of another bacterium is practically not expressed, so we doubt it plays a role here. We now point also towards a potential role of aggregation in KRV36 natural habitat in biofilm with cyanobacteria, in which the lower oxygen concentration might also play a role. Further genome mining and experiments on cell adhesion would be highly interesting but are beyond the scope of the current manuscript.

Editor (Comments for the Author):

The authors are invited to detail further their description of the identification of the strain KRV36 and of the metabolic pathway analysis in the methodological section.

response E: We added the missing information on programs used for assembly to the section on genome sequencing. The strain identification was based on the 16s rRNA gene retrieved from the completely sequenced genome that we submitted to NCBI as outlined in the section “phylogenetic analysis”. At this moment, we curated the KEGG pathways we were most interested in only manually as stated in the “genome sequencing” section. We intend to submit a full description of the strain KRV36 as a new *Sediminicoccus* species with an extended pathway analysis backed by physiological experiments in a separate manuscript and therefore do not wish to extend this part here.

On behalf of the authors

Yours Sincerely,

David Kaftan Ph.D.
Center of Excellence Algatech
Inst of Microbiology CAS, Trebon

Re: mSystems01311-23R1 (A photoheterotrophic bacterium from Iceland has adapted its photosynthetic machinery to the long days of polar summer)

Dear Dr. David Kaftan:

Your manuscript has been accepted, and I am forwarding it to the ASM production staff for publication. Your paper will first be checked to make sure all elements meet the technical requirements. ASM staff will contact you if anything needs to be revised before copyediting and production can begin. Otherwise, you will be notified when your proofs are ready to be viewed.

Featured Image Submissions: If you would like to submit a potential Featured Image, please email a file and a short legend to mSystems@asmusa.org. Please note that we can only consider images that (i) the authors created or own and (ii) have not been previously published. By submitting, you agree that the image can be used under the same terms as the published article. Image File requirements: TIF/EPS, 7.5 inches wide by 8.25 inches tall (at least 2,250 pixels wide by 2,475 pixels tall), minimum 300 dpi resolution (600 dpi preferred), RGB, and no figure elements, e.g., arrows or panel labels. The legend should be a short description of the image, 1-2 sentences recommended.

Sincerely,
Angela Re

Editor
mSystems

Reviewer #1 (Comments for the Author):

The authors have responded perfectly to the reviewers' suggestions and questions. I have no more suggestions. A small issue, the reference format in the current version seems not to be uniform. The author should check it more carefully, to meet the journal's requirements.

Reviewer #2 (Comments for the Author):

The authors have addressed all the raised questions. No further questions or comments.

Reviewer #3 (Comments for the Author):

The authors have already addressed all my questions. I have no additional comments.